Prediction of sentiment polarity in restaurant reviews using an ordinal regression approach based on evolutionary XGBoost

Al-Qudah Dana A. 1 2
Al-Zoubi Ala’ M. 2
http://orcid.org/0000-0002-1454-8822 Cristea Alexandra I. 3
Merelo-Guervós Juan J. 4
Castillo Pedro A. 4
Faris Hossam 1 5 hossam.faris@ju.edu.jo
1 King Abdullah II School for Information Technology, The University of Jordan , Amman , Jordan
2 Faculty of Information Technology, Applied Science Private University , Amman , Jordan
3 Department of Computer Science, Durham University , Durham , United Kingdom
4 Department of Computer Architecture and Technology, Universidad de Granada , Granada , Spain
5 Research Centre for Information and Communications Technologies of the University of Granada (CITIC-UGR) , Granada , Spain
Agapito Giuseppe
Electronic publication date: 2025 Jan 9
Publication date: 2025
Volume: 11
Electronic Location ID: e2370
Received 2021 Jul 23; Accepted 2024 Sep 9
Copyright: © 2025 Al-Qudah et al.
Copyright year: 2025
Copyright holder: Al-Qudah et al.
License: This is an open access article distributed under the terms of the Creative Commons Attribution License, which permits unrestricted use, distribution, reproduction and adaptation in any medium and for any purpose provided that it is properly attributed. For attribution, the original author(s), title, publication source (PeerJ Computer Science) and either DOI or URL of the article must be cited.
License URL: https://creativecommons.org/licenses/by/4.0/

Keywords: Ordinal regression, Sentiment polarity, Evolutionary, Particle swarm optimisation, XGBoost

Funding: Ministerio Español de Ciencia e Innovación (formerly known as Ministerio español de Economıa y Competitividad) PID2020-115570GB-C22 MCIN/AEI/10.13039/501100011033 and PID2023-147409NB-C21 MICIU/AEI/10.13039/501100011033 Deanship of Scientific Research, The University of Jordan This work is supported by the Ministerio Español de Ciencia e Innovación (formerly known as Ministerio español de Economıa y Competitividad) under project PID2020-115570GB-C22 MCIN/AEI/10.13039/501100011033 and PID2023-147409NB-C21 MICIU/AEI/10.13039/501100011033 and the Deanship of Scientific Research, The University of Jordan.

==============================
As the business world shifts to the web and tremendous amounts of data become available on multilingual mobile applications, new business and research challenges and opportunities have been explored. This research aims to intensify the usage of data analytics, machine learning, and sentiment analysis of textual data to classify customers’ reviews, feedback, and ratings of businesses in Jordan’s food and restaurant industry. The main methods used in this research were sentiment polarity (to address the challenges posed by businesses to automatically apply text analysis) and bio-metric techniques (to systematically identify users’ emotional states, so reviews can be thoroughly understood). The research was extended to deal with reviews in Arabic, dialectic Arabic, and English, with the main focus on the Arabic language, as the application examined (Talabat) is based in Jordan. Arabic and English reviews were collected from the application, and a new model was proposed to sentimentally analyze reviews. The proposed model has four main stages: data collection, data preparation, model building, and model evaluation. The main purpose of this research is to study the problem expressed above using a model of ordinal regression to overcome issues related to misclassification. Additionally, an automatic multi-language prediction approach for online restaurant reviews was proposed by combining the eXtreme gradient boosting (XGBoost) and particle swarm optimization (PSO) techniques for the ordinal regression of these reviews. The proposed PSO-XGB algorithm showed superior results when compared to support vector machine (SVM) and other optimization methods in terms of root mean square error (RMSE) for the English and Arabic datasets. Specifically, for the Arabic dataset, PSO-XGB achieved an RMSE value of 0.7722, whereas PSO-SVM achieved an RSME value of 0.9988.

Introduction

Social online eating, food ordering, and related applications and websites have become common in people’s lives (Roh & Park, 2019). However, while the food industry is growing, many businesses are closing down (Martín-Valdivia et al., 2013; Kauer & Moreira, 2016). Restaurants are often affected by their reputations and customers’ feedback (Vinodhini & Chandrasekaran, 2012). However, receiving useful feedback can be challenging; hence, online ordering apps and websites usually allow customers to directly provide feedback and ratings. This way, business owners can be frequently updated on their customers’ needs and opinions and make enhancements to their services accordingly. However, since manually tracking hundreds or thousands of feedback messages is almost impossible, feedback tracking remains challenging.

Currently, sentiment polarity (SP) is the recommended way to address this issue. SP is the process of dealing with written text in order to identify and interpret the opinions expressed in it. These opinions are usually categorized, for example, as positive, negative, or neutral (Krishna et al., 2019; AlZu’bi et al., 2022). The need to use SP is derived from the tremendous amount of data collected, extracted, loaded, and used in structured and unstructured text harvested from the internet (Ravi & Ravi, 2015). SP uses combined or separate natural language processing (NLP) approaches and various text analysis techniques so that the reviews and the opinions expressed therein can be classified based on various criteria (Krishna et al., 2019).

However, there is no single best procedure for applying SP, as it is a wide, developing area of computational text analysis. Researchers have used different performance evaluation methods, such as text classification, including naïve Bayes, decision tree classifiers, and N-fold cross-validation (Karsi, Zaim & El Alami, 2017; Adnan, Sarno & Sungkono, 2019). There are two main sentiment polarity approaches: knowledge-based or machine learning-based SP. The main difference between these two approaches is that the former is based on NLP algorithms or lexicon methods (Neethu & Rajasree, 2013), whereas the latter focuses on data polarity (i.e., whether a text is negative, positive, or neutral) based on training on data previously labeled by humans (Gautam & Yadav, 2014).

Arabic NLP is a relatively new research area, especially for dialectal Arabic. One such dialect that is particularly underexplored is the Jordanian dialect. Moreover, as Arab speakers are often multilingual and use more than one language in their daily lives, a multi-language approach is required. However, very few studies of this nature exist (Dashtipour et al., 2016). Talabat, a multilingual restaurant and food ordering application, was examined in this study to explore users’ reviews, feedback, and ratings of restaurants in Jordan.

As mentioned above, the dramatic expansion of application usage and the enormous availability of text-based data have raised many challenges for business owners and application developers. The main challenges are related to the need to analyze customer reviews and categorize them as positive, negative, or neutral. Each polarity helps business owners maintain their advantages and improve their shortcomings in different ways. Also, rating services require combined analysis with text reviews, as ratings alone do not give enough information for business owners to understand them. The sentiment underlying each review must be clear, and automated procedures should be provided for business owners to understand how reviews and ratings are labeled. These challenges force developers and researchers to find enhanced methods for searching for, learning about, and evaluating the sentiments and ratings of reviewers in both Arabic and English languages.

The misclassification problem between multi-class cases was addressed using a model based on ordinal regression formulation in this research. Ordinal regression can help determine the link between distinct classes, which is difficult to determine using other methods. This model was also proposed to cover all the matters discussed above. The model was construed as follows. The first phase was data description and collection, during which the dataset was built based on Arabic and English reviews and ratings collected from the Talabat application. In the next step (data preparation), processes such as stemming, cleansing, and rooting were conducted in both Arabic and English. The proposed approach was then explained thoroughly, the model was evaluated, and the main results were explained and interpreted.

This work addresses two main research questions: How can sentiment analysis methods and techniques improve the quality of feedback extracted from reviews and ratings in Arabic (with its dialects) and English?

Can the proposed method optimize the sentiments of feedback in Arabic and English compared to state-of-the-art methods?

In previous work, Al-Qudah et al. (2020) demonstrated the superiority of the eXtreme gradient boosting (XGBoost) algorithm in classic classification problems for sentiment analysis. The algorithm effectively predicted and analyzed customers’ opinions of an e-payment service when the researchers combined a neutrality detector model with XGBoost and a genetic algorithm to solve a classic multi-class problem. However, in the current study, XGBoost is applied to entirely different problems, namely ordinal regression in sentiment polarity. This research aims to address the challenges posed by businesses by analyzing online restaurant reviews in a multi-language environment.

The main contribution of this research is the proposal of an ordinal regression formulation to minimize the misclassification gap between multi-class problems. Moreover, it combines the swarm-intelligent optimizer with the powerful XGBoost algorithm to handle multi-class sentiment classification problems. The proposed approach has two primary advantages. Firstly, the swarm-intelligent optimizer automatically tunes the parameters of the XGBoost algorithm, eliminating the need for human intervention to complete this task. Secondly, it handles the classification problem as an ordinal multi-class classification problem, considering the importance of class order. One of the main features of the proposed data harvesting and cleaning approach is that feedback messages in both Arabic and English were harvested. Dialectical Arabic was also considered due to the lack of standardization for dialects and their sheer number. After data cleaning was performed for the dataset, a particle swarm optimization (PSO) algorithm was used to optimize the XGBoost algorithm and detect the five classes of sentiment polarity. Ordinal regression was also used to classify the data.

The questions answered in this study are as follows: How effective are current sentiment analysis methods in accurately identifying the sentiments in reviews? Does incorporating multi-language support, specifically for the Arabic language, improve the accuracy of sentiment analysis? Can the proposed method using ordinal regression and PSO-support vector machine (SVM) with the XGBoost algorithm increase the accuracy of sentiment classification? What specific challenges are faced in processing and analyzing sentiments in dialectal Arabic?

The main aims of the study are as follows: To study customers’ reviews, feedback, and ratings in Arabic in general and the dialectal Arabic used in Jordan using a popular restaurant ordering and reviewing application called Talabat.

To address the challenges faced by restaurant owners and application developers in effectively organizing and comprehending large amounts of available text-based data as feedback for their businesses.

To delve into customers’ feedback, ratings, and reviews, not as plain text data but in a way that enables the classification of the sentiments of feedback into positive, negative, and neutral categories.

To conduct a comprehensive analysis amalgamating the reviews and feedback with rating services, mixing two different types of data analysis—namely, text data and categorical data—to provide a holistic understanding of customer perceptions.

The remainder of the article is organized as follows. First, previous studies on ordinal regression sentiment polarity are introduced in “Related Work”. Then, the methods used in this study are described in “Preliminaries”. Next, the proposed approach is discussed in “Methodology”. The results of the experiments are subsequently analyzed in “Experiment and Results”. Finally, the conclusions and future directions are presented in “Conclusion”.

Related work

The food service industry is economically important, and many of its services have moved online. Therefore, much research has been conducted to improve the specific services provided to online users to make ordering and receiving food using these services easier and more satisfying. This research analyzes the text harvested from users’ feedback on such systems. Many researchers have conducted similar work that has enriched the state of the art. However, this work presents a new combination of algorithms that has not been sufficiently explored while considering both the Arabic and English languages in the Jordanian market specifically.

Sun, Guo & Zhu (2019) recently conducted a study on recommender systems of online Chinese restaurants following uncertainty theory and using sentiment polarity. They highlighted the importance of analyzing customer reviews, as the overall ratings did not accurately represent their reflections and opinions. Furthermore, they claimed that these reviews should be divided based on opinions’ similarities and the designs of the utilized recommender systems. They analyzed the text provided by users by determining the main attributes of the sentiment polarity. This was done by acquiring the main attributes from the reviews by performing a fine-grained classification, which eventually aids in assessing the polarity and strength of text. This part of the study was conducted using HowNet (Dong & Dong, 2006). The researchers continued their work by using the uncertainty theory to build the recommender algorithms and models. Finally, they used K-nearest neighbor and K-means algorithms to further classify and cluster reviews while also determining the accuracy of these recommender systems.

Adnan, Sarno & Sungkono (2019) analyzed restaurant reviews using a decision tree (J48) algorithm. They used users’ reviews of and comments about restaurants in Surabaya on TripAdvisor and harvested the data using web scraping software called WebHarvy, which saves data in Excel format. The data contained information such as the name of the customer; their rating, comment text, and comment title; and the name of the restaurant. The researchers also pre-processed the data using the Natural Language Toolkit available in Python. In the pre-processing stage, many operations were done, such as tokenization, slang word removal, stop word removal, and symbol removal. After the cleaning and data processing stages, the researchers calculated the number of appearances of each word in the text using a J48 decision tree. They obtained a precision of 48, a recall of 36.8, an accuracy of 45.6, and an F-measure of 41.4, indicating a classification recommendation to identify good restaurants.

Krishna et al. (2019) conducted more comprehensive research by studying various machine learning techniques to analyze the sentiments of restaurant reviews. They used a dataset in Tab Spaced Values format. They pre-processed the data by applying various cleaning and tokenization techniques. They then prepared bags of words created from pairs of different documents by performing disjoint unions of the words and summing up their multiplicities. Afterwards, they applied various classification algorithms, such as naïve Bayes, support vector machine (SVM), decision tree, and random forest. The experiment was run three times, and SVM had the highest prediction accuracy (94.56).

Zhang et al. (2015) focused on the type of sentiment analysis conducted, namely, whether the analysis is on a document-review level, a structure level, or a phased level. The results suggest that precision can reach 90% when comparing sentiment analysis results from document level or structure level. Meanwhile, precision can be around 70–80% when using phrase-level analysis.

From another perspective, sentiment polarity in text has also been explored using XGBoost, with researchers mainly analyzing tweets. This is because XGBoost performs well when applied to large-scale problems, as it employs highly flexible operations and calculations on most regression, classification, and ranking problems (Chen & Guestrin, 2016). XGBoost has been used in many studies related to sentiment polarity, such as the work of Jabreel & Moreno (2018). They used XGBoost to analyze text extracted from tweets, focusing on lexicons-based features. They compared this approach with a deep learning approach called N-Stream ConvNets. The main outcome of their research is that combining an ensemble technique with XGBoost helped improve the performance of the booster. The researchers suggested that this combination provides similar or better results compared to those given by the deep learning technique.

Furthermore, sentiment polarity was discussed by Kern et al. (2021), who studied sentiment analysis in German language using cluster analysis. The researchers used an unsupervised method to cluster the polarity of the German language, with the results showing that different German dictionaries were similar; the main differences were detected in the structure and outliers. They based their conclusions on a K = 3 cluster using K-means.

Meanwhile, Bhoi & Joshi (2018) discussed how to classify a certain aspect in a sentence. They argued that sentiment polarity should be used for classification not by looking at a sentence as a bag of words but rather at items in temporal order. The authors used common pre-processing steps before testing many algorithms, such as naïve Bayes, decision trees, support vector machine, random forest classifier, extra trees classifier, and XGBoost, combined with other deep learning techniques. Their results indicate that XGBoost performed well as a classifier and that other deep-learning algorithms could provide more accurate results.

Nobre & Neves (2019) studied sentiment in the financial market by combining principle component analysis (PCA), XGBoost, genetic algorithms, and discrete wavelet transform (DWT). PCA and DWT were combined into one system to accomplish high returns while minimizing possible risks. This combined approach yielded good results, with an average rate of return of 49.26 in the portfolio. In other work, Song et al. (2020) explored steel property optimization using both PSO and XGBoost. The authors attempted to build an optimization model with 27 features for machine learning to predict the tensile strength and plasticity. The complexity of the model alongside the numerous features was studied using the combined approach. The main results indicate that XGBoost was the most reliable prediction model examined in the study. The PSO approach aided the final optimization of the model for iron and steel production.

Le et al. (2019) combined the PSO and XGBoost methods and found that smart cities play an important role in the development of countries. They recorded higher reliability for their combined method than for other machine learning algorithms regarding the prediction and optimization of the heating loads of buildings. Although the work presented by these researchers targeted different applications, they proposed the combination of PSO and XGBoost, which the current work further explores in a different domain, in addition to ordinal regression algorithms, as explained further.

Huang et al. (2019) explored sentiment analysis on social media data using a novel approach. This approach primarily revolves around the integration of three distinct attention models for predicting sentiments. Their experiments showcase the efficacy of utilizing various types of datasets, including weekly labeled data and manually labeled datasets, alongside three different models tailored for sentiment classification. Notably, their methodology employed deep multi-modal fusion to discern features and establish correlations between visual and textual content. They employed a mixed fusion framework to merge sentiment analysis, emphasizing regional word features to differentiate between models. Their proposal encompassed models designed to effectively learn emotion classifiers for both visual and textual content.

Huang et al. (2020) work extends previous discussions by introducing a novel model known as the attention-based modularity gated network. This model determines correlations between image and text modalities while extracting discriminative features for multi-modal sentiment analysis. Specifically, the authors employed a visual semantic attention model to learn visually attended features for each word, effectively integrating sentiment information from both modalities. Additionally, they propose a long short-term memory (LSTM) network to adaptively learn multimodal features, selecting modalities that exhibit stronger sentiment signals. The model incorporates a self-attention mechanism to enhance semantic understanding.

Later work delved into multi-modal approaches for analyzing both visual and textual data (Thuseethan et al., 2020). This study highlights the potential pitfalls of blindly merging textual and image data for sentiment analysis and classification and proposes that the interrelations between multi-modal data should be explored using a deep association learner, which is adept at discerning relationships by leveraging learned visual and textual features, thereby automatically discriminating between features extracted from text and images. The two distinct streams of model features can then be extracted, focusing on the most pertinent aspects related to sentiment. Subsequently, sentiment estimation was performed through a late fusion mechanism. Comprehensive evaluations demonstrate promising results, indicating the capability to classify data based on sentiments, whether they comprise text, images, or a combination of both.

Naeem et al. (2022) implemented various machine learning models to gauge sentiment polarity in user reviews on the Internet Movie Database. The process involved preprocessing the reviews to eliminate noise and redundant information, followed by employing classification models such as support vector machines (SVMs), naïve Bayes, random forest, and gradient boosting. Feature engineering techniques, including term frequency-inverse document frequency (TF-IDF), a bag of words, global vectors for word representations, and Word2Vec were applied, alongside hyperparameter tuning, to enhance classification accuracy. The results reveal that SVM, when combined with TF-IDF features, achieved the highest accuracy of 89.55%. However, user sentiment contradictions pose a challenge to accurate classification. To address this, TextBlob was employed to assign sentiment labels to the review dataset. The results of TextBlob-assigned sentiments indicate a potential accuracy of 92% when using the proposed model.

Another study showed that sentiment expressed in tweets regarding deep fakes holds considerable importance in understanding public perception (Rupapara et al., 2021). This study introduced a deep learning approach to assess the sentiment polarity of such tweets and proposed a stacked bi-directional long short-term memory (SBi-LSTM) network for sentiment classification. Additionally, various traditional machine learning classifiers, such as support vector machine, logistic regression, Gaussian naïve Bayes, extra tree classifier, and AdaBoost classifier, were explored alongside feature extraction methods like term frequency-inverse document frequency, and bag of words. The performances of deep learning models, including long short-term memory network, gated recurrent unit, bi-directional LSTM, and convolutional neural network+LSTM, were also evaluated. The findings demonstrate that the SBi-LSTM model outperformed both traditional machine learning and deep learning approaches, achieving an accuracy of 92%.

Rupapara et al. (2021) conducted another study in the same context by detecting fake news, which has become a crucial area of research, particularly in languages like Urdu, which is spoken by over 230 million people, but for which investigations remain limited. This study evaluated the effectiveness of various machine learning classifiers and a deep learning model in detecting fake news in Urdu. Classifiers such as logistic regression, support vector machine, random forest, naïve Bayes, gradient boosting, and passive aggression were employed, alongside analysis of term frequency-inverse document frequency and bag of words features. Based on a dataset of 900 manually collected news articles, the study found that random forest performed the best, achieving an accuracy of 0.92 with bag of words features. In comparison, machine learning models generally outperformed deep learning models such as long short-term memory and multi-layer perceptron in this context.

The research also discussed that as restaurants have joined online platforms like UberEATS, Menulog, or Deliveroo, customer reviews have become vital for assessing company performance (Adak, Pradhan & Shukla, 2022). Food delivery service (FDS) organizations seek to utilize customer feedback to address complaints and enhance customer satisfaction. This study examines machine learning (ML), deep learning (DL), and explainable artificial intelligence (XAI) methods for predicting customer sentiments in the FDS sector. A review of the existing literature highlights the prevalence of lexicon-based and ML techniques for sentiment prediction based on customer reviews, with a limited adoption of DL due to interpretability concerns. Key findings reveal that many models lack interpretability, posing challenges for organizational trust. While DL models offer high accuracy, they lack explainability. However, this issue can be addressed using XAI techniques. Future research should focus on integrating DL models into FDS sentiment analysis and incorporating XAI methods to enhance model explainability.

Shahi, Sitaula & Paudel (2022) focused on Nepali-language COVID-19-related tweets and proposed an analysis of sentiments using both syntactical and semantic information. They combined TF-IDF and FastText text representation methods to create hybrid features for enhanced discrimination. Nine widely used machine learning classifiers were implemented based on three feature rep- representation methods: TF-IDF, FastText, and a hybrid method. The methods were evaluated using a NepCov19Tweets dataset, with data categorized into positive, negative, and neutral classes. The results indicate that the hybrid feature extraction method outperformed individual methods across nine machine learning algorithms, demonstrating superior performance compared to state-of-the-art techniques.

Ordinal regression has been shown to be effective when working with labeled and unlabeled data. Rafique et al. (2022) focused on understanding the accuracy of transductive ordinal regression as a highly accurate algorithm when applied to pre-processed data. This type of algorithm is well-developed and is proposed to work prominently with sentimentally labeled data in the Bulgarian language. In addition, Kapukaranov & Nakov (2015) studied movie reviews in Bulgarian using ordinal regression. They used multiple classification algorithms to set a threshold region to classify positive, negative, and neutral reviews. The aim was to predict which of these regions would be set to be positive, negative, and neutral. They divided the regions based on the sentiments of the reviews.

Meanwhile, Loke et al. (2020) studied sentiment polarity using neural network architecture. They studied attention-based sentiment analysis and used neural networks to classify the outcomes. They used F1 scores, accuracy loss function, and receiver operating characteristic (ROC) area under the curve (AUC) to evaluate the results. The results indicate F1 scores of around 72%, accuracy of 92%, and losses of around 20.

Moreover, Saad & Yang (2019) examined the sentiment polarity of Twitter data (tweets) using different machine learning algorithms to deal with ordinal regression problems. The results indicate highly accurate prediction rates in comparison to other algorithms such as SVM, random forest (RF), and decision trees (DT). The researchers combined the ordinal regression classifier with XGBoost and PSO.

Al-Qudah et al. (2020) employed XGBoost with the genetic algorithm for parameter optimization in a classification problem. The researchers predicted the sentiments (positive, negative, or neutral) of e-payment service reviews in the Arabic language. Accordingly, in the present article, XGBoost was implemented in order to handle ordinal regression in sentiment polarity (ratings of 1, 2, 3, 4, and 5) for a multi-language problem. Furthermore, PSO was used to optimize the XGBoost parameters, thereby enhancing its performance. Unlike the previous work, this study handled ordinal regression for the sentiment polarity issue while investigating two different languages.

This approach differs from the previous works by combining PSO and XGBoost to optimize parameters for the ordinal regression problem for multilingual restaurant reviews. The selected languages involve different pre-processing techniques and require more analysis to deal with. Furthermore, XGBoost was compared with various metaheuristic algorithms, including PSO, the well-known whale optimization algorithm (WOA) (Mirjalili & Lewis, 2016) and multi-verse optimization (MVO) (Mirjalili, Mirjalili & Hatamlou, 2016). Then, the best algorithm was chosen for another comparison with support vector machine, another well-known classifier.

Based on the extensive study of the recent literature and case studies, a comprehensive understanding of the existing work in the field has been obtained. The main contributions of our work, in comparison to the available studies, are categorized into three main domains: data source, modeling techniques, and optimization. Most of the data used in previous studies were collected from social media—for example, Twitter (now known as X) and Facebook. Moreover, in one case, the data source was a delivery service platform. Our data source is one of the most important and widely used food ordering applications, whose business model includes reviews of the delivery service model. In addition, this study examines the usage of ordinal regression to overcome issues related to misclassification. Such issues have not been thoroughly discussed in the literature as they relate to the Arabic language or while considering the dialectical Arabic used in Jordan. The final contribution of this study relates to the optimization of the utilized models; few articles have discussed optimization in sentimental analysis. Meanwhile, our research utilized two main optimizers.

Preliminaries

Ordinal regression/classification

Ordinal classification or regression is a type of multiclass classification method for classes that have an ordering relationship, even though they do not have any meaningful numeric differences (Gaudette & Japkowicz, 2009). This problem is considered one of the most significant tasks in relation-learning. Ordinal data are frequently classified in scenarios involving human-made scale problems. In other words, classes may contain different sizes of values such as small, medium, and large (Yıldırım, Birant & Birant, 2019) or cheap, normal, and expensive. Meanwhile, ordinal categorical variables can be factors or predictors in several statistical procedures, like linear regression.

The purpose of ordinal data prediction is to determine how to calculate distances between categories without knowing how to calculate distances between variables. These issues lay between the categorization and regression techniques utilized in psychology, sociology, and other disciplines. The modeling of human preference levels (e.g., on a scale from 1–5, depending on how strong one’s preference is) is an example of ordinal regression (Bürkner & Vuorre, 2019). One of the advantages of ordinal measurement is that it simplifies the processes of collecting and categorizing data.

Class labels convey information about the order of classes; an average class vector has a higher (or better) rating than a poor class, but a good class exceeds both. Two factors are significant in this type of problem. First, different types of misclassification have various consequences; for example, misclassifying an excellent teacher by erroneously categorizing him as poor is more severe than incorrectly classifying him as very good. Second, more accurate models may be built by using ordering information.

Ordinal classification has been applied to several real-world problems, including problems related to medical sciences (Cardoso, da Costa & Cardoso, 2005), collaborative filtering (Shashua & Levin, 2003), information retrieval (Herbrich, Graepel & Obermayer, 1999), econometric modeling (Mathieson, 1996).

XGBoost algorithm

The XGBoost algorithm is an enhanced version of the gradient boosting algorithm that applies decision trees. XGBoost was developed by Chen & Guestrin (2016) to solve regression and classification problems efficiently and rapidly. It was developed to improve machines’ speed and exploit the full functionality of their resources, such as memory and hardware. The XGBoost algorithm is valued owing to its ability to reduce time consumption while handling, problems such as missing values, parallel execution, and the use of the optimal machine resources. Furthermore, the algorithm adopts the regularization technique to decrease and prevent the overfitting problem. Further extensions, including stochastic and gradient boosting, further improve its performance (Song et al., 2020).

As the base learners are tree algorithms, the XGBoost algorithm divides the dataset attributes into conditional nodes consisting of several branches and a leaf node (Chen et al., 2020). Moreover, the hyperparameters of the XGBoost algorithm are considered quite important in seeking the optimal results for a specific problem; therefore, tuning these parameters is necessary.

PSO algorithm

Particle swarm optimization (PSO) is a well-known metaheuristic algorithm inspired by the behavior of flock birds (Kennedy & Eberhart, 1995). The population-based algorithm simulates the position of these birds in order to achieve an optimal solution. In PSO, particles are denoted as a set of solutions called a population, and the solution consists of different parameters found in a given multidimensional space. These particles (i.e., the population) are grouped to perform a swarm that searches the space at a specified velocity to find the optimal solution.

Particles can save their memory to keep tracking the former best position (solution). The positions of these particles can be adjusted until the optimal solution is discovered according to the personal best experience (pbest) and global best, which represents other members’ best experiences (gbest) (Ding, Zhou & Bi, 2020). Moreover, the historical behavior of particles and their neighbors helps update the particles’ velocities while searching the search space (flying). Therefore, the search process tends to improve in every iteration (Ghamisi & Benediktsson, 2014).

In this work, PSO was combined with XGBoost to optimize the ordinal classification of restaurant reviews by finding the optimal parameters for XGBoost.

Methodology

This section presents the methodology for detecting restaurant reviews. Four phases are described in detail: data description and collection, data preparation, the proposed approach, and model evaluation. In the first phase, the data information, the source of the data, the method used for the collection process, and the statistics details are described. In the second phase, the data are prepared. This phase consists of formatting, cleaning, stemming, and feature extraction. The third phase presents design issues, the fitness function, and the system architecture of the proposed approach. The last phase is the evaluation phase.

As for business applications, using reviews of customers to evaluate and enhance the service provided is vital. Thus, extra focus is given to studying user satisfaction using data mining algorithms, namely XGBoost combined with PSO for ordinal regression.

Data description and collection

The research targets a specialized food ordering application called Talabat (Talabat, 2004), which is well-known in the Middle East. It operates in countries such as Saudi Arabia, Oman, Kuwait, Qatar, the UAE, and Jordan. It works as a mediator between registered restaurants and customers. This research analyzed 2,000 reviews annotated with their ratings from consumers who used Talabat. The ratings are divided into five classes, from 1 to 5, with one being the lowest rating and five being the highest. All opinions associated with the ratings were harvested using a customized Python script.

Services cannot be improved without the appropriate feedback from stakeholders, regular customers, or both. Therefore, such reviews, if acted upon, can make a service better in different ways that owners—including restaurant owners—may have neglected.

Problems can be avoided by analyzing the customers’ reviews, as well as the ratings they use, which express their sentiments. Automated procedures, which save time and money, are enticing even for small businesses. Therefore, a polarity analysis is performed in the form of a rating prediction on bi-lingual data.

The reviews in this article were collected from the Talabat website using a crawler tool. Each review contains a customer username, the date of the review, and the associated rating. The ratings were crawled as images of 1, 2, 3, 4, and 5 stars. The reviews were written in two languages: Arabic and English.

The data consist of 1,927 instances and 1,292 features (terms). More details on the dataset can be found in Table 1.

Table 1 Bilingual dataset description.

Dataset	Instances	Features	Classes	
English	935	385	5	
Arabic	992	707	5	

Data preparation

In this stage, the reviews (data) went through several standard pre-processing procedures (Faris et al., 2017; Habib et al., 2018). In other words, these procedures prepared the data so it could be read by the classifiers. The procedures included formatting the data, removing missing values, and cleaning (Hassonah et al., 2020).

The labeling process was not needed for customer reviews since the reviews had already been labeled by the customers. However, to enhance accuracy, several experts were asked to read a sample of the data and ensure that the labeling was correct.

The first step in the data preparation process was to split the data into two separate datasets based on language (English and Arabic). This had to be done due to the differences in these languages’ characteristics, namely in terms of stop words, stemming processes, prefixes, and suffixes.

For the Arabic dataset, various stop words were removed so that the true meanings of sentences could be identified. Moreover, a normalization technique was applied to discard special characters and non-Arabic letters to reduce the number of extracted terms. Finally, a stemming process was performed in order to decrease the duplication of extraction terms (prefixes and suffixes). After different stemming methods were tested, the Arabic light stemmer method was chosen for the Arabic language (Al Ameed et al., 2005).

Meanwhile, for the English dataset, stop words such as “the,” “and,” and “but” were removed. Then, special characters and symbols were removed to decrease the number of terms that could be extracted. Furthermore, the snowball stemmer was used to remove the suffixes and prefixes of the text (Porter, 2001).

After the previous steps were completed, a tokenization procedure was applied to break the words into tokens by analyzing the text linguistically and splitting the words. The term frequency-inverse document frequency (TF–IDF) was used to extract features (Ghag & Shah, 2014). This technique calculates the relevance numerically for a particular document, as shown in Eqs. (1) and (2). TF(t,d) is the number of words ( t) in the document ( d) and IDF(t) is the inverse document frequency.

(1) IDF(t)=log(NDF(t))

The TF−IDF value can be computed as follows.

(2) TF.IDF(d,t)=TF(d,t)×IDF(t)

In Eq. (1), N denotes the number of all documents and DF(t) is the number of documents in which the words ( t) occur.

Eventually, the tokenization process (TF-IDF) was followed to convert the data into a matrix in order to run it through the classifier. However, some problems, including missing values and noisy data, could have been encountered and needed to be resolved before the data were examined. These problems can be solved by majority vote (Xia et al., 2017) and normalization (Wang et al., 2006).

Proposed approach

This subsection describes the proposed approach applied to the datasets. Furthermore, the sentiment identification model is assessed via a root mean square error (RMSE) and mean absolute error (MAE). The model is displayed in Fig. 1 below.

Figure 1 General overview of the applied framework.

After the datasets were prepared, the classification model was run on the data. The XGBoost algorithm was the main classifier used and was compared with various state-of-the-art algorithms for the task of sentiment polarity prediction. Recently, XGBoost has gained attention in the literature, especially in machine learning applications (Al-Qudah et al., 2020). Nevertheless, because it has many parameters, it is challenging to find the appropriate combinations between applications and associated problems (Jiang et al., 2019).

Thus, particle swarm optimization (PSO) was applied, as it can help find the optimal combination of hyper-parameters for XGBoost. It is also faster and more accurate than the other methods described in the literature (Lin et al., 2008), such as grid search. The grid search algorithm can search for the best parameters in a given range for different models. However, it takes a substantial amount of time to run and find the local optimum solution. Using PSO is more efficient for such problems. Meta-heuristic algorithms (Al-Zoubi et al., 2019), like PSO, have three main components: searching, learning, and evaluation.

XGBoost has many hyper-parameters. Those that are the most frequently used in the literature were selected (Table 2).

Table 2 The XGBoost parameters used and their descriptions, ranges, and best values (Al-Qudah et al., 2020; Jiang et al., 2019).

#	Parameters	Description	Range	
1	min_child_weight	A leaf’s minimum weight	[1–20]	
2	gamma γ	Reduction minimum loss required for the partition	[0.1–5.0]	
3	subsample	Ratio of training records	[0.1–1.0]	
4	colsample_bytree	Features sub-sample ratio	[0.1–1.0]	
5	max_depth	Tree depth maximum	[1–20]	
6	learning_rate	Step size	0.02	
7	n_estimators	Number of trees employed	1,000	

Design issues

Two design issues should be considered when an optimization algorithm is applied to a problem: the design representation for the solution and the fitness function (Fig. 1).

Solution representation: The PSO swarms are designed to represent the solution to the problem. In this work, the main problem relates to finding the optimal parameters for XGBoost. The swarms consist of a one-dimensional set of randomly generated numbers corresponding to the parameter’s value. These generated numbers are scaled from 0 to 1 to simplify the selection criteria of the parameters, as shown in the following equation:

(3) B=A−minAmaxA−minA(maxB−minB)+minB

where A denotes the value that needs to be scaled, B represents the new scaled value, minA is the lower bound, and maxA is the upper bound of the old range. The lower and upper bounds of the new range are denoted by minB and maxB, respectively.

Fitness function: An evaluation criteria was applied to improve the generated solutions from PSO, and feedback from XGBoost was provided for every iteration. The root mean square error (RMSE) was selected as a fitness function. RMSE is the most common metric used for ordinal regression problems in the literature (Li, Wang & Dey, 2019; Gaudette & Japkowicz, 2009; Shi et al., 2018) due to its ability to show the degree of deviation between predicted and original labels. Therefore, the PSO algorithm was modified to minimize the fitness value (fitness function). The equation used to calculate RMSE is given below:

(4) RMSE=1nΣi=1n(yi−y^i)2

where n is the total number of samples yi is the actual value, and y^i is the estimated value.

System architecture: In this phase, the datasets were split into a training and a testing set using the 10-fold splitting criteria (Shao et al., 2013; Basiri, Ghasem-Aghaee & Aghdam, 2008). The dataset was divided into k parts, with the training set containing k−(1/k) parts and the test set containing the remaining (1/k) parts. This procedure ensured that the training and testing sets were differentiated and that the optimal model was attained (Hassonah et al., 2020), which is useful, especially in cases when the training data is limited.

In the first iteration, PSO generated a random set of real numbers in a vector form. Then, XGBoost started the training process using the parameters selected by PSO. After the training was completed, XGBoost sent the fitness value to PSO. These steps were then repeated until the termination criteria was reached—in this case, the maximum number of iterations. Consequently, the best-selected values generated by PSO were used in the testing phase. All previous steps were repeated k times, and the average value was recorded.

Evaluation

Several evaluation measures were applied to assess and calculate the model’s performance. In addition to the root mean square error (RMSE), the mean absolute error (MAE) measure was been used; it was calculated using the following equation:

(5) MAE=1n∑i=1n|yi−y^i|

where n is the total number of samples, yi is the actual value, and y^i denotes the estimated output value.

Furthermore, an extended analyzed evaluation process was applied in order to state the errors in the predicted ordinal classes by using a confusion matrix table.

The classes that were correctly predicted were labeled as true positive (TP), while false negative (FN) was used to refer to classes that were wrongly predicted as incorrect ratings. Similarly, false positive (FP) represents classes that should have been predicted as wrong ratings but were predicted as correct ratings. Finally, true negative (TN) denotes classes that were correctly predicted as wrong ratings.

Experimentation and results

This section describes the results of several experiments applied to the prepared datasets. The results show that the combination of XGBoost with a metaheuristic (PSO) obtained better results than each method independently. Both datasets (Arabic and English) were processed in the same stages.

First, classical machine learning methods (as explained below) were compared with the default XGBoost algorithm. Next, in the second phase, the proposed PSO-XGB was compared to other metaheuristic algorithms. In the final phase, the PSO-XGB and a classification model combined with the best metaheuristic were compared. Two evaluation measures were applied to these models, namely, root mean square error (RMSE) and mean square error (MAE).

All experiments were conducted on an Intel Core i5-6400 personal computer with 8 GB RAM. The proposed model was implemented on Python 3.7.

First, XGBoost was applied to the English and Arabic datasets and compared with classic machine learning models (J48, RF, KNN, and naïve Bayes (NB)). Then, PSO and other metaheuristic algorithms (WOA and MVO) were combined with XGBoost and compared. Furthermore, a feature importance analysis was carried out. Finally, PSO was combined with another classification model (SVM) (Ala’M et al., 2020) and compared with the proposed PSO-XGB model. Moreover, a detailed examination was conducted to state the errors of the model using a confusion matrix. All experiments adhered to the 10-fold criteria.

Phase 1: Comparison with classic machine learning models

In this phase, the default XGBoost algorithm was applied to the English dataset and compared to classic machine learning models.

As shown in Table 3, XGBoost achieved the best RMSE result of 1.2472. The next-best result was obtained by RF (2.7373). As for the MAE measure, NB yielded the best result of 0.9613, followed by XGB (0.9925).

Table 3 Results for the English dataset.

Algorithm	RMSE	MAE	
J48	3.0384	1.4916	
RF	2.7373	1.1362	
KNN	2.9898	1.0547	
NB	2.9725	0.9613	
XGB	1.2472	0.9925	

Regarding the Arabic dataset, Table 4 shows that the best result was obtained by XGBoost (0.9259), while the second-best result was yielded by RF (1.9750). In terms of MAE, XGBoost also achieved the best result (0.4167), followed by NB (0.4876).

Table 4 Results for the Arabic dataset.

Algorithm	RMSE	MAE	
J48	2.2233	0.8396	
RF	1.9750	0.7018	
KNN	2.1893	0.6534	
NB	2.1611	0.4876	
XGB	0.9259	0.4167	

This outcome confirms that XGBoost performs better than the other classifiers; thus, it was used in the next phase. Specifically, XGBoost outperformed the second-best algorithm in the English and Arabic datasets with RMSE values of 1.4901 and 1.0491, respectively. XGBoost excels due to its effective implementation of stochastic gradient boosting as well as its inbuilt regularization tools that prevent overfitting.

Phase 2: Comparison with different metaheuristic algorithms combined with the XGB

Several metaheuristic algorithms were compared after being combined with XGB in order to identify the best combination. Three algorithms were used in this phase, namely, PSO, WOA, and MVO, each of which has shown excellent results in the literature for different problems (Ala’M et al., 2018; Abd Elaziz et al., 2019; Rostami et al., 2020).

As shown in Table 5, PSO-XGB produced the best RMSE of 1.0993, while the MVO-XGB achieved the best-second result (1.1168) and WOA-XGB yielded the worst result (1.1188). In terms of MAE, PSO-XGB also obtained the best result (0.9258), followed by MVO-XGB (1.0520) and WOA-XGB (1.0390).

Table 5 Results for PSO, WOA, and MVO on the English dataset.

Algorithm	RMSE	MAE	
PSO-XGB	1.0993	0.9528	
WOA-XGB	1.1188	1.0390	
MVO-XGB	1.1168	1.0520	

Moreover, PSO-XGB showed better performance than the other algorithms in both measures, indicating that it is the best algorithm for this problem for the English dataset. The convergence of the three algorithms can be seen in Fig. 2.

Figure 2 Convergence for PSO-XGB, MVO-XGB and WOA-XGB on the English dataset.

Regarding the best values of the XGB parameters, the PSO-XGB (the superior model) selection can be found in Table 6, where, for example, the min_child_weight is equal to 2 and γ is equal to 3.42. The rest of the values are listed in the table.

Table 6 Best parameters for the English dataset.

Parameters	Best value	
min_child_weight	2	
gamma γ	3.42	
subsample	0.96	
colsample_bytree	1	
max_depth	2	
learning_rate	0.91	
n_estimators	10	

As can be seen, the metaheuristic algorithms’ selection of the best parameters enhanced the results in all measures when compared with the first phase. This shows that the XGBoost parameters had a huge impact on its performance. Therefore, this problem requires the use of metaheuristic algorithms.

As with the English dataset, three algorithms (PSO-XGB, WOA-XGB, and MVO-XGB) were compared for the Arabic dataset. As shown in Table 7, the lowest RMSE result (0.7858) was obtained by the PSO-XGB algorithm. The second-lowest value was achieved by the MVO-XGB algorithm (0.8896); this is unlike the English dataset, for which this algorithm produced the worst result. For the MAE measure, the best result was obtained also by the PSO-XGB algorithm (0.3999), followed by the GB-MVO and WOA-XGB algorithms. The convergence of the three algorithms can be seen in Fig. 3. As mentioned earlier, the metaheuristic algorithms also improved the results for both measures since the best parameters were selected for XGBoost.

Table 7 Results for PSO, WOA, and MVO on the Arabic dataset.

Eng	RMSE	MAE	
PSO-XGB	0.7722	0.3999	
WOA-XGB	0.8531	0.4717	
MVO-XGB	0.8186	0.4377	

Figure 3 Convergence for PSO-XGB, MVO-XGB and WOA-XGB on the Arabic dataset.

The best values of the XGBoost parameters that were selected by PSO-XGB can be found in Table 8.

Table 8 Best parameters for the Arabic dataset.

Parameters	Best value	
min_child_weight	1	
gamma γ	5	
subsample	1	
colsample_bytree	0.54	
max_depth	12	
learning_rate	0.72	
n_estimators	510	

Feature importance analysis

Additional analyses were performed to identify the most important features (words). These keywords are considered the top influencing features for each dataset in predicting the ratings of the reviews. Feature importance was performed using the XGBoost algorithm. The mechanism employed for this task was calculated and weighted by the number of times the feature appeared in the tree structure (Manju, Harish & Prajwal, 2019). The number of split points for each attribute responsible for improving the performance measure was used to determine the weights of features. These split points were defined by the Gini index method, and the average of all features’ importance was calculated across all trees in the model.

Figure 4 illustrates the top 20 features or words for both datasets. These features directly indicate consumers’ opinions. Some words show similarity in importance, while others depict different views or thoughts.

Figure 4 Top 20 features for English and Arabic datasets.

The first five features for the Arabic dataset are F32, F69, F356, F124, and F224. Their translations are “taste,” “prices,” “employee,” “a lot,” and “great,” respectively. The first feature (taste) indicates how important the sense of taste is to consumers and the extent to which the taste of food is considered an essential factor when choosing between restaurants. The second feature (prices) demonstrates the cost of the meal and if such a price is justified (high or low). The third feature (employee) indicates satisfaction with the service provided bystaff. The fourth feature (a lot) does not suggest any exact meaning other than the amount of something. The fifth feature (great) implies the adjective’s positive meaning but without specifying which characteristics it refers to. Nevertheless, such a feature (great) is used by consumers when they emphasize their opinions, whether their review is positive or negative.

Meanwhile, the top five features for the English dataset are “price,” “service,” “bad,” “great,” and “late.” The most important feature was the price, as the cost is crucial to consumers when they judge a meal or restaurant. Concerning the second feature, the service betokens how much can be significant to rate a restaurant. This indicates that excellent or horrible service is considered critical in the selection process. The third and fourth features both denote the quality of either the price, food, or service of a restaurant. Similar to the “great” feature, “bad” emphasizes the sentiment of the review. Regarding the fifth feature, receiving the food late is considered important, especially when a customer gives a low rating.

Moreover, both datasets show similarities and differences in the order of the features. Features such as “price,” “taste,” “late,” “service,” “cold,” and “delivery” are similar, whereas “expectations,” “overrated,” and “spicy” show differences.

Phase 3: different classifiers combined with the best metaheuristic algorithms

In this final phase, PSO-XGB was compared with a well-known classifier (SVM) and combined with the PSO algorithm. As can be seen in Table 9, PSO-XGB achieved the best performance in terms of RMSE.

Table 9 Results for XGB and SVM combined with PSO for the English dataset.

Algo	RMSE	
PSO-XGB	1.0993	
PSO-SVM	1.4204	

In this final phase, PSO-XGB was compared against a well-known classifier (SVM) in the literature and combined with the PSO algorithm. As can be noticed in Table 9, PSO-XGB achieved the best performance in terms of RMSE.

The Arabic dataset results are shown in Table 10. The PSO-XGB algorithm outperformed the PSO-SVM algorithm in terms of RMSE.

Table 10 Results for XGB and SVM combined with PSO for the Arabic dataset.

Algo	RMSE	
PSO-XGB	0.7722	
PSO-SVM	0.9988	

Furthermore, the 10-fold results in Table 11 for both datasets using PSO-XGB demonstrate significant improvements. Additionally, the statistical test (p-value) indicates that, in comparison with PSO-SVM, the results for PSO-XGB are very small, suggesting strong statistical significance (Table 12). For instance, if the p-value is 0.05 or less, it confirms that the observed differences are not due to random chance but reflect a real improvement in performance.

Table 11 The 10-fold results of the PSO-XGB for both English and Arabic datasets (italic results are the best results).

10-Folds	English	Arabic	
	PSO-XGB	
Fold 1	1.1100	0.7853	
Fold 2	1.1155	0.7905	
Fold 3	1.1309	0.8051	
Fold 4	1.1256	0.8153	
Fold 5	1.0993	0.8004	
Fold 6	1.1401	0.7950	
Fold 7	1.1452	0.7801	
Fold 8	1.1208	0.7722	
Fold 9	1.1354	0.8100	
Fold 10	1.1500	0.8202	
Average	1.12728	0.79741	

Table 12 Statistical test (p-value) comparing PSO-XGB with PSO-SVM.

Data	PSO-SVM	
Enlgish	1.83E-04	
Arabic	2.13E-34	

Furthermore, the 10-fold results found in Table 11 for both datasets using PSO-XGB demonstrate significant improvements, with the best results highlighted in italic font.

Analysis of the error for PSO-XGB

Figures 5 and 6 show the errors of the predicted ordinal classes through the confusion matrix. These errors explain the difference between each class and the class adjacent to it.

Figure 5 Confusion matrix values for the English dataset.

Figure 6 Confusion matrix values for the Arabic dataset.

The data in Fig. 6 (Arabic language) show better performance in terms of sentiment ordinal regression prediction than Fig. 5 (English language). In other words, for example, instances like rating (1) and (2) classified (error) as rating (5) in the English language, unlike the Arabic language, where there is less error in this matter. Conversely, 19 instances were classified as a rating of 1 when they should have been given a rating of 5. Again, this kind of error is less common in the Arabic data (Fig. 6).

After an extensive review of these errors (English data), it was determined that they occur for several reasons (ranked according to the majority), including: Irony and sarcasm. Some users use positive words to describe negative opinions. For example, one person had a rough day and expected to eat delicious food from a restaurant; however, he did not like the food. His review, which accompanied a (1) rating, stated, “That’s just what I needed today!”

Word ambiguity and choosing the wrong words to describe something (occurred when reviewers left reviews in a language other than their native language).

Incorrect selection of the rating (occurred if the rater was confused about the meanings of ratings).

The findings indicate that it is hard to capture such reviews in the NLP model. However, this extended analysis explains the precise nature of such errors. Therefore, on such websites, restaurant owners should be more careful when dealing with English reviews, as the reviewers are non-native English speakers.

Moreover, the accuracies for the confusion matrices of the PSO-XGBoost algorithm (Figs. 5 and 6) are 0.970 and 0.982 for the English and Arabic datasets, respectively. However, such results are not relevant for this kind of problem (ordinal regression), since the weight of misclassification is not detected.

Conclusion

This work presented an ordinal regression sentiment polarity approach using the PSO-XGBoost algorithm to assess restaurant reviews. Two types of pre-processing procedures were handled—one for each language dataset (Arabic and English). Furthermore, the PSO algorithm functioned as an identifier and optimization technique for the XGBoost parameters; it determined the optimal combination and eventually yielded the best possible performance. It obtained superior results while handling complex tasks such as ordinal regression problems (e.g., restaurant reviews).

The proposed approach was compared with other methods in three phases: first with standard classifiers (J48, RF, KNN, NB), then with other recent metaheuristic algorithms (MVO and WOA), and finally with SVM. The proposed approach achieved better results than other methods in all phases. More specifically, within the English dataset, PSO-XGB achieved an RMSE of 1.0993, outperforming WOA-XGB (1.1188), MOV-XGB (1.1168), and PSO-SVM (1.420). Regarding the Arabic dataset, the proposed PSO-XGB yielded an RMSE of 0.7722, meaning it outperformed WOA-XGB (0.8531), MOV-XGB (0.8186), and PSO-SVM (0.9988).

Regarding the research questions, the proposed method can assist restaurant owners and provide early alerts and feedback, allowing owners to focus on the most important terms (features) without having to read all reviews in both languages. This, in turn, enables them to make better business decisions by utilizing relevant information. The method also reminds business owners to be cautious when handling non-native English speakers’ reviewers. Further, the work achieved advanced performance using the evolutionary XGBoost criteria, which performed better than state-of-the-art criteria.

Future research should implement more sophisticated model that can capture systematic reviews. In addition, sampling more data would allow more terms and features to be mapped with labels based on the tree classification algorithm in XGBoost. Moreover, attention should be paid to detecting irony and sarcasm when assessing reviews. Doing so would ensure a comprehensive understanding of reviewers’ true sentiments and help avoid misinterpretations that could impact decision-making and customer satisfaction. As a result, the study’s domain knowledge will improve, allowing it to be applied to various fields, including predictive text and other approaches to natural language processing. Finally, more comparisons with other algorithms can be conducted to compare the running times of various measures.

Supplemental Information

Supplemental Information 1 XGBoost.

Supplemental Information 2 PSO-SVM Code.

Additional Information and Declarations

Competing Interests

Author Contributions

Data Availability

The authors declare that they have no competing interests.

Dana A. Al-Qudah conceived and designed the experiments, performed the experiments, analyzed the data, prepared figures and/or tables, and approved the final draft.

Ala’ M. Al-Zoubi conceived and designed the experiments, performed the experiments, performed the computation work, prepared figures and/or tables, and approved the final draft.

Alexandra I. Cristea conceived and designed the experiments, analyzed the data, prepared figures and/or tables, authored or reviewed drafts of the article, and approved the final draft.

Juan J. Merelo-Guervós conceived and designed the experiments, prepared figures and/or tables, authored or reviewed drafts of the article, and approved the final draft.

Pedro A. Castillo conceived and designed the experiments, prepared figures and/or tables, authored or reviewed drafts of the article, and approved the final draft.

Hossam Faris conceived and designed the experiments, performed the experiments, analyzed the data, performed the computation work, authored or reviewed drafts of the article, and approved the final draft.

The following information was supplied regarding data availability:

The code is available in the Supplemental Files.

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
