# Peer review of "Prediction of sentiment polarity in restaurant reviews using an ordinal regression approach based on evolutionary XGBoost"

_PeerJ Computer Science, doi:10.7717/peerj-cs.2370_

## Round 0.1 · original submission · Major Revisions

The authors are requested to revise the manuscript as per the suggestions given by the reviewers.

·

Basic reporting

• Some minor English proofread is still needed. Here are some examples:
Line 50: “…is the current recommended way the way to address this”
Line 76: “Then the a best algorithm…”
Line 370-371: “Three algorithms used in this phase, namely, PSO, WOA, and MVO. Each of which have shown…”

• Please indicate in parenthesis the meaning of RMSE in the abstract, as well as the first time it is used in the prose (line 280). Take this advice in effect for every other acronym used along the paper.

• It should be clarified from the abstract whether the proposal deals with intermingled Arabic and English in the same text, or their experimentation is conducted separately for each language (line 36).

• Avoid paragraphs that are too long, as it makes reading cumbersome (example: lines 142-157; lines 158-170).

• There is a typo on line 87: “ecause of the importance of this industry…".

• Although it is not a review article, it is good practice to briefly explain how the results presented in the Related Work have been produced. These search strings can enrich the Related Work section and it is left to the discretion of the authors to use them:

https://scholar.google.com/scholar?hl=es&as_sdt=1%2C5&as_ylo=2017&as_vis=1&q=intitle%3A%22sentiment+polarity%22+intitle%3A%22review%22+%2B%22prediction%22&btnG=

https://scholar.google.com/scholar?q=intitle:%22sentiment+polarity%22+intitle:%22review%22+%2B%22prediction%22&hl=es&as_sdt=1,5&as_vis=1

• Consider changing Figure 5 into a Table, as this looks more appropriate for the kind of visual display used.

• Center the equations (1), (2), (3), (4), (5).

• Rename section “EXPERIMENT AND RESULTS” TO “EXPERIMENTATION AND RESULTS” (line 340).

• Conclusions must start with capitals: “sentiment polarity is…” (line 452).

Experimental design

• Methods are described with sufficient detail, in order to replicate, explaining each of four phases: Data description and collection, Data preparation, Proposed approach, and Evaluation.

• The journal requests in its guidelines that the research questions be well defined, relevant and meaningful. Please state in the Introduction what the research question is, which could be along the lines of:

"What approach is most recommendable to predict sentiment polarity in restaurant reviews written in Arabic?". The answer to this question must be made explicit in the Conclusions.

Validity of the findings

• The research findings appear to be valid. Nevertheless, it is advisable to give numerical results an interpretation. In lines 386-387 the authors state that “This proves that XGBoost performs better than the other classifiers, which is why we will be using it in the next phase”. Please explain the magnitude of this performance improvement. Is it good, acceptable, too good? Why is this so?

• Underlying data have been provided in a clear and exhaustive fashion, which is commendable in this type of research.

Additional comments

• The authors' research is valuable, as it presents a problem little addressed in Arabic languages. Despite not being a management science paper, this reviewer considers it a good idea to explain from the Introduction how managers are going to improve decision-making thanks to the use of their proposal, just to better inform a wider audience. One single paragraph would suffice.

Reviewer 2 ·

Basic reporting

1. the manuscript is clearly suffering from lack of professional, ambiguous language structure.

2. In related work need to point out the strength & weaknesses of each paper, to identify the design gaps, and problem areas, which should help define a problem statement for this research
3. third person grammar is preferred to first person in all formal documents, like this paper. - We ,Our.

4. All symbols and variables should be in Italic fonts; vectors and matrices should be in Bold font.

5. Please check all the instances of "et.al" and change them to the appropriate form in italics.

Experimental design

1. It is always recommended to give the key contributions of the paper at the end introduction
2. all acronyms should be fully expanded the first time they’re used in any document – not arbitrarily, as seen here in page 6 RMSE
3. In Methodology for data preparation and proposed work, a simple illustrative description would help clarify the concepts. These would be easier for readers, and less writing for authors, if a simple illustration is also shown. These are unclear and confusing; would be helpful to have a simple block diagram at least, to explain the concepts.
4. The main components of Abstract section should be topic of interest/discussion, main outstanding problems, or gaps/deficiencies, proposed solution to resolve them, high level brief on how the proposed theory/design was verified, key results/findings, plans/recommendations for future research. All formal documentation like this paper, should be in third-person English grammar instead of the first person [use of I, we, our, etc.]. Better to quantify instead of qualitative assessment. Acronyms and abbreviations should be avoided in Abstract unless unavoidable.
5.Introduction needs to clearly present the topic domain, including key positive and negative or unpreferred aspects. Challenges mentioned need to be clarified from which a problem statement could be derived and defined. This is followed by presentation of a proposed solution – theory or model (point wise), ….followed by verification strategy and anticipated results or success.
6. There are many Undefined terms are seen throughout the document. such as in line no:102 'HowNet', 107 : WebHarvy , fitness value etc
7.We also need to include critical notes on the strengths and weaknesses reviewed in the literature. We can then use these to define a problem statement and propose a solution

Validity of the findings

1. The figure 2 must be moved to proposed methods.
2.We should explain in brief, the functionality or significance of equation, rather than merely presenting the equations. Prior to this we should also ensure all symbols, variables are defined aptly
3. In experiments and results section, how should the presented scores be interpreted is unknown and unspecified. There should be a paragraph or two, on discussion of the results, including any surprises or lessons learned from this analysis. Just doing comparison and getting good results will not work. You need to answer the questions: Why and how.
4.Prior to jumping into results, we need to present an overview of the test plan(ratio) for verification of the proposed model in this paper.
5.Conclusion should mainly reaffirm accomplishment of all objectives of this research, stating only the key/best [& worst, if any] parts of the findings. Also can include key lessons learned, which should suggest plans/recommendations for future research, and how this study advanced the domain knowledge.
6. Table 11 and 12 should be be a confusion matrix figure which should have a figure ID and a precise caption. Need a legend that explains all the symbols, color codes, and abbreviations
7. There is no accuracy and f1-score found in the paper. The values of RMSE and MAE neither seem attractive nor worth the effort – we need to highlight the qualitative aspects of the accomplishments/findings to justify this research.

Additional comments

Major revisions required

---

## Round 0.2 · Major Revisions

Dear Authors,

Please revise the manuscript based on the reviewers comments.

The reviewers have requested that you cite specific references. You may add them if you believe they are especially relevant. However, I do not expect you to include these citations, and if you do not include them, this will not influence my decision.

Also, improve the English language presentation of this manuscript.

Thank you.

Reviewer 2 ·

Basic reporting

The authors have responded to most of our comments but there are still some minor comments that require addressing.
- There are some acronyms used without being defined
- This research requires minor check of the reference list.
- English Writing: This research requires moderate proofreading of the entirety of the research to eliminate all the issues related to spelling, typos and grammar mistakes.

Experimental design

All comments addressed

Validity of the findings

All comments addressed

·

Basic reporting

This paper presents an ordinal regression-based sentiment polarity prediction. Following are some points to further improve the paper;

1. The literature could be expanded to multimodal sentiment analysis as it is the current trend. For instance, authors can include the below papers;
Image–text sentiment analysis via deep multimodal attentive fusion, published in Knowledge-based Systems
Multimodal deep learning framework for sentiment analysis from text-image web Data, published in Wi-IAT
Attention-based modality-gated networks for image-text sentiment analysis, published in ACM Transactions on Multimedia Computing

2. Providing a brief description of Figure 1 would be beneficial to readers.

Experimental design

Very clear and complete!

Validity of the findings

Very clear and complete!

·

Basic reporting

Dear Author,
Your work is good and the presentation of data is also good. please consider my comments to further improvements in the manuscript.
1- Add key points of the study in the introduction section with bullets.
2- Add recent studies in the literature review related to sentiments and also add a tabular summary of the literature.
"Classification of movie reviews using term frequency-inverse document frequency and optimized machine learning algorithms"
Deepfake tweets classification using stacked Bi-LSTM and words embedding
"Comparative analysis of machine learning methods to detect fake news in an Urdu language corpus"
Sentiment Analysis and Emotion Detection on Cryptocurrency-Related Tweets Using Ensemble LSTM-GRU Model
Inquest of Current Situation in Afghanistan Under Taliban Rule Using Sentiment Analysis and Volume Analysis
Learning sentiment analysis for accessibility user reviews
Review prognosis system to predict employees job satisfaction using deep neural network
3- Please compared your proposed approach for sentiment analysis with other methods.
please develop the mentioned study methods and test your data.
COVID-19 Vaccination-Related Sentiments Analysis: A Case Study Using Worldwide Twitter Dataset
Deepfake tweets classification using stacked Bi-LSTM and words embedding
Tweets classification on the basis of sentiments for US airline companies
4-Please add k-fold results.
5- Likewise, I do not see a clear hypothesis from the study.
6- If possible, please proofread the manuscript carefully in case of grammatical mistakes.
7- The conclusions should be more relevant according to the results achieved.
8- Perform statistical T-test please check mentioned study.
Minimizing the overlapping degree to improve class-imbalanced learning under sparse feature selection: application to fraud detection

Experimental design

No comment

Validity of the findings

No comment

Additional comments

No comment

Reviewer 5 ·

Basic reporting

N/A

Experimental design

N/A

Validity of the findings

N/A

Additional comments

Author attempted to present the resturent review classificaiton usign ordinal regression. Also, the comments from previous round are addressed in some extent. However, the work still needs major revision. I have following feedback.

Comments
1. The abstract needs a rewrite so that it reflects the main contribution of the work along with some comparative data.
2. I couldn’t find the code for PSO-XGBOOST, it seems only PSO-SVM code is there.
3. The flow of the paper is not smooth in the introduction part. The author must summarize the existing research, their gap and the contribution of the proposed work in this section more precisely.
4. The related work section can be extended with the following recent work on sentiment analysis at the discretion of the author.
a. A Hybrid Feature Extraction Method for Nepali COVID-19-Related Tweets Classification
b. Sentiment Analysis of Customer Reviews of Food Delivery Services Using Deep Learning and Explainable Artificial Intelligence: Systematic Review
c. Deep learning-based methods for sentiment analysis on Nepali covid-19-related tweets
5. The research gap in existing literature needs to be mentioned explicitly while summarizing the existing work. Better if the performance of existing work is summarised in the form of a table.
6. Why ordinal regression is best suited for this restaurant review classification problem? Why the normal multi-class classification method is not suitable? How does ordinal regression contribute to a more accurate model? What are the measurement criteria for such a model?
7. What are the benefits of PSO over the grid search to find the optimal parameters? It can be seen that grid search is widely used in the existing literature. The comparison of run-time between grid search and PSO needs to be presented to validate the authors' reasoning.
8. How author ensure that the customer labels the review accurately? There might be some bias in such data annotation process.
9. What do R2, R3, and so on represented in fig 1? Please follow the similar notation for all ratings.
10. What is “algo” in table 3. Please follow the standard notation.
11. Why classification accuracy of the proposed algorithms are not compared? If the authors presented it as a regression problem, the coefficient of determination should be used as a measure.

---

## Round 0.3 · Minor Revisions

Based on the reports from our reviewers and my own assessment as Editor, I am pleased to inform you that your manuscript is almost acceptable for publication in PeerJ Computer Science, provided that you address some essential revisions suggested by our reviewer ("the validation of results through k-fold and statistical test is still missing").

Please submit a revised manuscript along with a point-by-point response letter. This letter should include detailed responses to each reviewer and editorial comment, specifying the exact amendments made to the manuscript and their locations

Additionally, ensure that your revised manuscript adheres to the journal style, as outlined in the Submission Guidelines on the journal homepage.

Furthermore, professional English editing may benefit the manuscript for greater clarity.

Sincerely,

Editor, PeerJ Computer Science

Reviewer 5 ·

Basic reporting

Still there are few typos and need moderate english editing

Experimental design

All comments addressed

Validity of the findings

still statistical test are missing

Additional comments

The author addessed most of the comments raised in the previous round. However, the validation of results through k-fold and statistical test is still missing.

---

## Round 0.4 · accepted · Accept

Based on the reviewers' reports, and my own assessment as Editor, I am pleased to inform you that the manuscript is acceptable for publication in PeerJ Computer Science.

·

Basic reporting

I accept the level of basic reporting in the revised paper as it meets the minimum standards required for clarity and comprehension.

Experimental design

I accept the experiment design in the revised paper as it demonstrates an adequate approach to addressing the research questions.

Validity of the findings

I accept the validity of the findings in the revised paper, as they are supported by appropriate analysis and evidence.

Additional comments

Good work

Reviewer 5 ·

Basic reporting

Good

Experimental design

Good

Validity of the findings

Good

Additional comments

All comments are addressed by the authors. Congratulations!